# Characteristics, Impacts and Trends of Urban Transportation

**Yuan Gao [1,2,*] and Jiaxing Zhu [1]**

1   School of Politics and Public Administration, Zhengzhou University, Zhengzhou 450001, China; 202012032010688@gs.zzu.edu.cn
2   Curtin University Sustainability Policy Institute, Curtin University, Bentley, WA 6102, Australia
*   Correspondence: yuan.gao@zzu.edu.cn

**Definition:** Economic growth, urban development and the prosperity of the automobile industry have driven a huge shift in global urban transportation from walking to public transportation and then to automobiles. Private mobility has become an important part of residents' daily trips. Cities, especially automobile-dependent cities, face a variety of negative impacts such as increased commuting distances, higher congestion costs, traffic accidents, traffic pollution including climate change, etc. Therefore, how to balance the relationship between people's growing demand for private motorization with the development of urbanization, modernization and motorization and the huge economic, social and environmental costs brought about by them, so as to realize the sustainable development of cities and transportation, is the main problem facing cities around the world. The entry focuses on trends in the sustainable development of urban transportation such as restrictions in private car ownership and use, electrification of urban transportation, intelligent transportation systems (including shared mobility, customized buses and Mobility as a Service/MaaS) and transit-oriented development (TOD). China, as the largest global automobile producer and consumer, represents and leads the growth and evolution of other emerging countries.

**Keywords:** urban transportation; transportation demand management; electrification of urban transportation; intelligent transportation system; transit-oriented development (TOD)

## 1. Background

Cities not only drive economic and value creation and play a vital role in many key social issues but are also the main link between human and environmental systems. In 1800, the global urbanization rate was only 2%, and it reached 50.16% in 2007 for the first time. It is predicted to be as high as 68.36% in 2050 [1]. Every year, more than 20 million people worldwide move from rural to urban areas, equivalent to the entire population of Romania in 2020. Although cities account for only 3% of the world's land area, they generate about 80% of the world's gross domestic product (GDP). The role of the city, especially in the new economic era, is becoming more and more important. It is not only the center of production, consumption, finance and service but also the center of innovation. The sustainable development of cities is crucial to achieving global sustainability within the Earth's environmental capacity.

The function and development of cities are inseparable from urban transportation. As a carrier to realize the movement of people and goods, urban transportation has gradually changed from a supporting facility of urban development to an important means of regulating the urban development model. Cesare Marchetti [2] initially found that the average daily trip time of residents, called the travel time budget, is about 65–70 mins based on research in major cities around the world. This is the "Marchetti constant". Beyond that amount, passengers perceive time spent traveling to and from their destination (usually work) as wasted or less valuable. According to the "one-hour travel circle", Newman and Kenworthy [3] divided cities into the categories of "Walking City" (prehistoric to 1850s), "Transit City" (1850s–1950s) and "Automobile City" (1950s-present). Currently, most cities

contain elements of all three urban forms, differentiated by different modal splits and urban densities. Even some urban centers still retain features of walking and transit urban fabrics, in the United States, the "Automobile City" is a typical model, for example: as of 2013, about 86% of Americans completed their commute by private car, and 76% of them drove alone [4].

## 2. Definition and Characteristics of Urban Private Transportation

### 2.1. Definition of Urban Private Transportation

Urban transportation is generally divided into three categories: public transportation, private transportation and cargo transportation. The car is regarded to have emerged in 1886 when Karl Benz patented his Benz Motorwagen and entered mass production in the 1920s. Private motor vehicles provide mobility and convenience, as well as on-demand, punctual and private transportation. They are also seen as symbols of social status and wealth. Generally, the private ownership of motor vehicles (including cars and motorcycles) is related to urban development, economic growth (including per capita income), urban form, public policies, the prosperity of the automotive industry, public transport system services and culture, etc. To analyze the contributors is not only helpful for predicting future private car ownership and use but also for forecasting private-car-related social and environmental issues such as energy consumption and Greenhouse Gas (GHG) emissions.

### 2.2. Characteristics of Urban Private Transportation around the World

Since the 1940s, automobile dependence (that is, a city prioritizing car services in urban infrastructure and urban planning due to the heavy use of private cars) has been considered by many developed cities to be conducive to promoting economic growth and increasing government revenue. Compared to European cities and developed Asian sample cities, it is found that the per capita private car ownership and usage, and the infrastructure to support private transportation (such as the freeway length and CBD parking spaces, etc.) in American, Canadian and Australian cities are far above the world average. The proportion of residents' daily trips by private car in such a city can be as high as 70–80%, making it a typical "automobile city".

A 2009 study by the Brookings Institute found that vehicle-kilometers traveled (VKT) nationwide peaked in 2004 and declined in 2007 for the first time since 1980s. Per capita VKT has a similar pattern [5]. Comparable stories have been found in Australian cities: growth has flattened and then started declining since 2004 [6].

The link between urban economic performance and the high mobility brought about by cars has been greatly weakened, and there is even a new phenomenon that is trending in the opposite direction, namely "Peak Car". Nemwan and Kenwothy [7] conducted research on 41 developed cities around the world and found that the average growth rate of VKT per capita in sample cities was 42% from 1960 to 1970; 26% from 1970 to 1980; and 23% from 1980 to 1990. It was only 7.2% between 1995 and 2005, less than one-third of what it was in the 1980s and less than one-sixth of what it was in the 1960s. This decoupling of car use from economic growth is also in line with the theme of UNEP International Resource Panel report *Decoupling Natural Resource Use and Environmental Impacts from Economic Growth*, suggesting that it is entirely possible to decouple wealth from fossil fuels [8].

### 2.3. Characteristics of Urban Private Transportation in China

There have been more than 45 million new urban dwellers per year in emerging economies, which means that urban areas, especially in developing countries, will grow rapidly. As the world's largest emerging economy, China has also experienced rapid urban development. Its total economic output was less than 5% of the world's total when New China was founded in 1949. However, it has surpassed Japan to become the second largest economy after the United States since 2010. Its urbanization rate was only 10.6% in 1949, with 132 cities, but the rate reached 60.6% in 2020, with 663 cities [9], equivalent to adding a new megacity every year. The area occupied by urban spaces in China also increased

from 6720 square kilometers in 1981 to 58,300 square kilometers in 2019, an increase of 7.68 times [10]. Meanwhile, China overtook the United States to become the world's largest auto producer and consumer in 2009, just 53 years after the establishment of the Chinese automotive industry in 1956 [11]. Economic growth, urban development and a boom in the auto industry have driven a dramatic shift from bicycles to cars as the primary mode of transportation in Chinese cities.

As shown in Figure 1, China's private car ownership has increased from 4 vehicles per thousand people in 2002 to 158 vehicles per thousand people in 2020, but the annual growth rate has dropped from 36.89% to 7.7%. In 2019, there were 2 cities with more than 5 million cars, 5 with more than 4 million cars, 11 with more than 3 million cars and 30 with more than 2 million cars. The spatial heterogeneity of vehicle ownership densities means that vehicle growth in developed cities may gradually slow, especially under the influence of license plate control policies; by contrast, inland Chinese cities are likely to continue their strong motorization momentum.

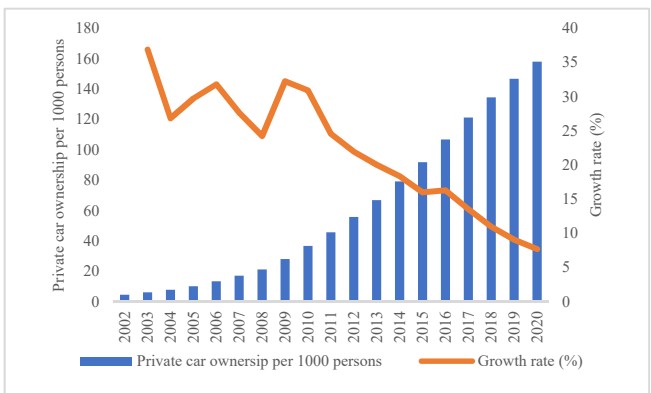 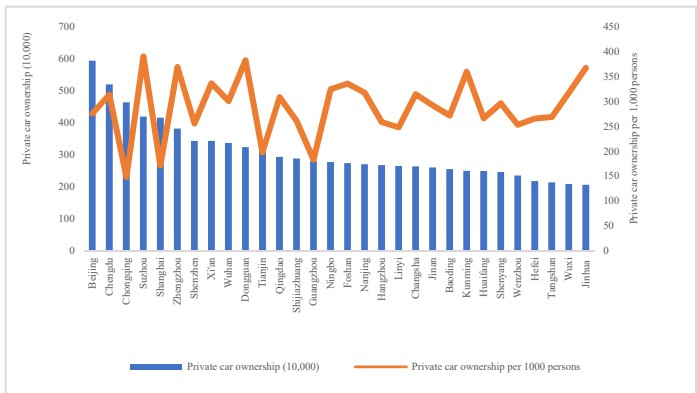

**Figure 1.** Private car ownership per 1000 persons and growth rate (%) in China from 2002 to 2020 (**left**) cities with more than 2 million cars owned in 2019 (**right**) Source: data compiled from National Bureau of Statistics, 2021 [9].

However, the number of private cars owned per 1000 people in various regions of the country is still far below the level of car ownership in cities in more developed countries. For example, in 2005–2006, the average number of cars per 1000 people in Australian, American, Canadian and European cities was 647, 640, 522 and 463, respectively [7]. The Chinese rate of car ownership is even far below those of countries such as Swaziland, El Salvador, Honduras, Guyana and Azerbaijan.

## 3. Definitions and Characteristics of Urban Public Transportation

Urban public transportation includes regular buses, bus rapid transit/BRT, light rail, subways, ferries, etc. Personal transportation includes personal motorized transportation (such as cars, motorcycles, electric vehicles, etc.), bicycles and walking. The two global oil crises in 1973 and 1979 led to high fossil fuel prices and tight supplies, thus promoting the development of public transportation systems. As an essential part of the urban passenger transportation system, urban public transportation has played a positive role in alleviating traffic congestion, reducing the demand for fossil fuels and environmental pollution, and ensuring the accessibility of cities for urban residents (especially urban vulnerable groups including the disabled, the elderly, etc.), thus aiding the sustainable development of cities and urban transport [12].

### 3.1. Regular Bus

3.1.1. Definitions of Regular Bus

The bus is intended to carry more passengers than private motor vehicles, with a scheduled routine, timeframe and other services to meet their daily travel purposes. It is

designated to provide an affordable, accessible and safe means of transportation to most local citizens. It is also regarded as a green transport mode due to its comparatively low energy consumption and exhaust emissions per passenger. With the availability of popular private transport modes and emerging rail services, passengers' dissatisfaction with bus services (personnel behavior, bus stop design, network coverage, etc.) has raised more attention, and the patronage of buses is accordingly declining both in absolute value and percentage. Recently, a bus renaissance has arisen due to improved bus service, emerging customized bus services and different attitudes of younger generations who treat riding a bus as a way to relax, communicate and reduce air pollution. Compared to rail and BRT, buses—especially community buses—provide first- and last-mile connections to other transport modes.

The trolley bus, which was first officially put into operation in 1901 in Germany, has also experienced decline and revival. Compared to regular buses, it generates lower levels of noise and air pollution and is thus regarded as a very important part of green public transport modes, especially when it is powered with renewable energy. It is also more flexible and more affordable compared to rail transport. However, it is less competitive in speed without dedicated lanes and more vulnerable to breakdowns of parts such as dual overhead wires. Currently, trolley buses with in-motion charging (IMC) are higher in capacity, more flexible, more efficient and produce zero emissions. This feature also reduces the construction and maintenance of charging poles.

### 3.1.2. Characteristics of Regular Bus around the World

At present, buses are the main form of public transportation in the world. The bus types used globally include the bi-articulated bus, articulated bus, standard bus, midibus and minibus. Standard buses account for 68% of buses worldwide [13]. Present urban forms have evolved from walking city fabric (daily activities are concentrated in the city core and can be completed within walking distance), transit city fabric (daily activities are concentrated in areas adjacent to the city core and can be completed within transit distance) and automobile city fabric (daily activities are concentrated in external areas far from the city core and can be completed by using automobiles). Most cities have a mixture of these fabrics but in different proportions. Hence, there are regional differences in the types and distributions of buses used in cities to meet daily necessities.

### 3.1.3. Characteristics of Regular Bus in China

The Five-Year Plan (FYP) aims to make plans for economic growth and social development at the national level in China. The Seventh FYP (1986–1990) proposed that the auto sector should be regarded as a pillar industry of the national economy to meet transportation demand. The Ninth FYP (1996–2000) determined that the auto industry, which performs as an engine of economic growth, needs to achieve mass production. The Tenth FYP (2001–2005) officially proposed the concept of "Encouraging Passenger Cars into Family". The automotive industry has gradually entered into prosperity, and private cars have correspondingly replaced public transport as daily transport modes. As shown in Figure 2, the number of buses (trolley buses included) and the operational length of buses have both experienced continuous growth. However, the total patronage of buses increased from 1996 and peaked in 2014, then decreased until recently, their share of total public transport peaked around the mid of the 2000s and then declined sharply. This continuous decrease is partly due to the increases in rail transport.

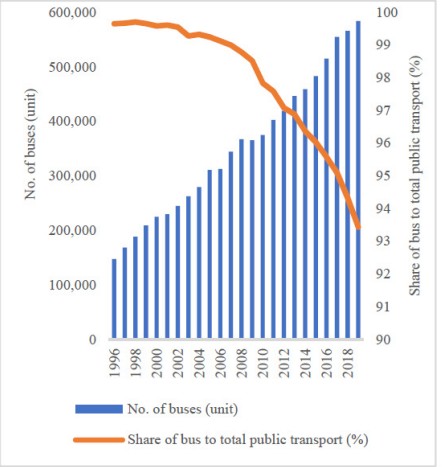 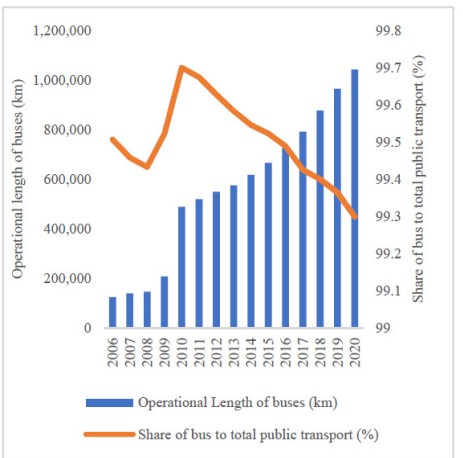 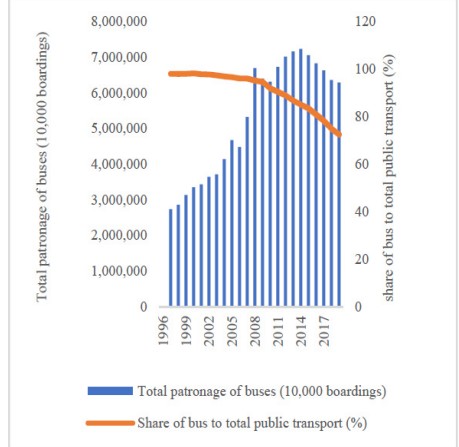

**Figure 2.** Trends of buses (trolley bus included) in China from 1996 to 2019 Source: data compiled from National Bureau of Statistics, 2020.

### 3.2. Bus Rapid Transit (BRT)

#### 3.2.1. Definitions of BRT

The term BRT was first used in a 1966 American Automobile Association/AAA study [14]. The term "Buses of High Level of Service/BHLS" is preferred in Europe. The world's first BRT system was developed in Curitiba, Brazil, in the 1970s. There are many definitions of BRT. Jaime Lerner, then the mayor of Curitiba, Brazil, believed that BRT is a "Surface Metro", that is, an urban road public transport service with the high service performance of a subway but cost similar to buses [15]. BRTs can be driven on dedicated lanes, in the middle or shoulders of existing roads or mixed with other motorized traffic (sometimes with signal priority at intersections) [16]. Depending on whether the BRT has a dedicated lane in the middle of the road (which can minimize conflicts with turning vehicles) or a dedicated station (passengers can buy tickets and level boarding before entering the station, etc.), the Institute of Transportation & Development Policy (ITDP) has divided BRT into BRT and BRT Lite (the latter being without fully segregated bus lanes and in some cases without dedicated stations).

#### 3.2.2. Characteristics of BRT around the World

As shown in Table 1, 179 cities around the world have opened BRT lines, with a total operating length of 5403 km and an average daily patronage of 33,751,575 persons as of September 2021 [17]. Among them, 59 cities in Latin America have opened BRT lines, with a total operating length of 1914 km, 35.41% of the global total; 45 cities in Asia have opened BRT lines, with a total operating length of 1691 km, 31.29% of the world total; 44 cities in Europe have opened BRT lines, with a total operating length of 875 km, only about half that of Asia, 16.19% of the world. Five cities each in Africa and Oceania have opened BRT lines, with operating lengths of 131 km and 109 km, accounting for 2.44% and 2.01% of the global total, respectively. In terms of average daily patronage, Latin America ranks first in the world with 20,938,474 persons, accounting for 62.17% of the world total. Among them, Brazil has an average of 10,796,415 daily riders due to its well-developed BRT system, 32% of the global total, more than half of the entire Latin American total and more than the Asian total. It is also three times that of Africa, Europe, Oceania and North America combined. The average daily patronage in Asia was 9,238,060 persons, accounting for 27.37% of the global total.

**Table 1.** Comparative analysis of global BRT by region in 2021.

| | Daily Average Patronage (Persons) | Global Share (%) | Number of Cities with BRT | Global Share (%) | Operation Length (km) | Global Share (%) |
|---|---|---|---|---|---|---|
| Latin America | 20,916,474 | 62.09 | 57 | 32.2 | 1886 | 35.07 |
| Asia | 9,238,060 | 27.42 | 45 | 25.42 | 1691 | 31.46 |
| # Mainland China | 4,375,250 | 12.99 | 33 | 18.64 | 672 | 12.50 |
| Europe | 1,613,580 | 4.79 | 44 | 24.85 | 875 | 16.28 |
| North America | 988,683 | 2.93 | 21 | 11.86 | 683 | 12.7 |
| Africa | 491,578 | 1.45 | 5 | 2.82 | 131 | 2.44 |
| Oceania | 436,200 | 1.29 | 5 | 2.82 | 109 | 2.02 |
| Worldwide | 33,684,575 | 99.97 | 177 | 99.97 | 5375 | 99.97 |

Source: data compiled from Global BRT data, 2021 [17].

However, as far as the global BRT system is concerned, the service it provides only accounts for 2.2% of all public transport trips and 0.3% of all motorized trips. Hence, there is still much room for development [18].

### 3.2.3. Characteristics of BRT in China

Kunming built China's first central bus lane in 1999; however, there is no dedicated BRT vehicle and station. Therefore, Beijing BRT, which started commercial operation in December 2004 and became fully operational in December 2005, is the first real BRT system in China. In 2008, China ushered in the first round of BRT outbreaks. As of September 2021, 38 cities with BRT systems in China have a total operating mileage of 672 km and average daily patronage of 4,375,250 boardings. Among them, the Guangzhou BRT system, which opened in 2010, consists of 30+1 (ferry lines) BRT bus lines. It is the BRT system with the largest single-line passenger flow in China and even in Asia.

### *3.3. Urban Rail Transit*

### 3.3.1. Definition of Urban Rail Transit

According to Common Terms of Urban Public Transport, urban rail transit is defined as "the general term for fast and large-capacity public transportation that is usually powered by electric energy and adopts wheel-rail operation". In China, it can be divided into trams, metro, light rail transit, suburban railways, single-rail transit, new transportation systems and maglev.

### 3.3.2. Characteristics of Urban Rail Transit around the World

The United Kingdom opened the world's first subway—the London Underground—in 1863. Since Asia began to invest in subways on a large scale in the 1970s, global rail transportation has developed tremendously, especially in China and India. By the end of 2020, a total of 538 cities in 77 countries and regions around the world had opened urban rail transit systems, with a total operating length of 33,346.37 km. Europe ranks first, accounting for 48.89% of the global total with a length of 16,302.33 km. Asia ranks second, accounting for about 39.36% of the global total with a length of 13,126.06 km. North America (2434.31 km), South America (779.8 km), Africa (416.37 km) and Oceania (287.5 km) together account for 11.75% of the global total.

In terms of patronage, the Asia-Pacific region reaches 26,960 million people per year, nearly half of the global total; it is followed by European cities, which account for about 1/5 of the global total. Since 2000, seven cities in Latin America have opened metro systems, marking the fastest growth in such systems since the 20th century. As of 2017, 19 countries in Latin America had opened metro systems with a total of 943 km of operating length, accounting for only 6.79% of the global total but providing 11% of the global passenger capacity. Among them, the five busiest lines (Mexico City in Mexico, Santiago in Chile,

Sao Paulo in Brazil, Caracas in Venezuela and Buenos Aires in Argentina) carry 76% of the passengers in Latin America. This also proves the potential growth capacity of the region. The bottom two regions for urban rail transit are North America and the Middle East and North Africa.

### 3.3.3. Characteristics of Urban Rail Transit in China

Beijing has opened the first Metro system in China in 1969. It took 38 years for China to create its first 1000 km of urban rail transit system. However, it took less than 5 years to create the second 1000 km. From then on, it increased by 1000 km every 1–2 years. As of the end of 2020, a total of 45 cities in mainland China had opened urban rail transit systems, with an operating mileage of 7978.19 km, accounting for 60.78% of Asia and 23.93% of the world total, respectively.

## 4. Impacts from Urban Transportation

The popularity of motor vehicles, especially private motor vehicles, has undoubtedly contributed to economic growth and urban development. However, it also has very clear negative effects on the economy, society and the environment, such as urban sprawl, increased long commutes, traffic congestion and accidents, etc. Energy consumption and GHG emissions are the most pressing issues caused by motor vehicles for urban sustainability.

### 4.1. Increase in Commuting Distance

Urban expansion, population agglomeration and automobile popularization have driven great changes in socioeconomic and spatial structures. Cities have gradually transformed from monocentric to polycentric, which has also led to higher levels of car use, more trips and longer commuting distances (or longer commuting times) [19]. Long-distance commuting has become the predominant type of daily commute in large cities of many developed countries [20]. The trend is also increasing in developing countries, especially in their megacities, which will lead to additional traffic, energy consumption and air pollution, and even traffic inequities for vulnerable groups.

### 4.2. Increase in Traffic Accidents

The number of deaths caused by road traffic accidents is as high as 3000 people every day, and it is the eighth leading cause of death in the world. The annual global loss is about USD 518 billion, accounting for about 3% of the GDP of the relevant countries [21]. The high-income countries with 40% of global vehicles account for 7% of the world's road traffic deaths. The low- and middle-income countries with only 60% of the world's vehicles account for 93% of the world's road fatalities. In terms of countries, the road traffic fatality rate continues declining rapidly in Japan and Russia while fluctuating in the United States, but all are lower than the global average. China's rate, although declining slowly, was still slightly higher than the global average in 2019 (see Figure 3) [22].

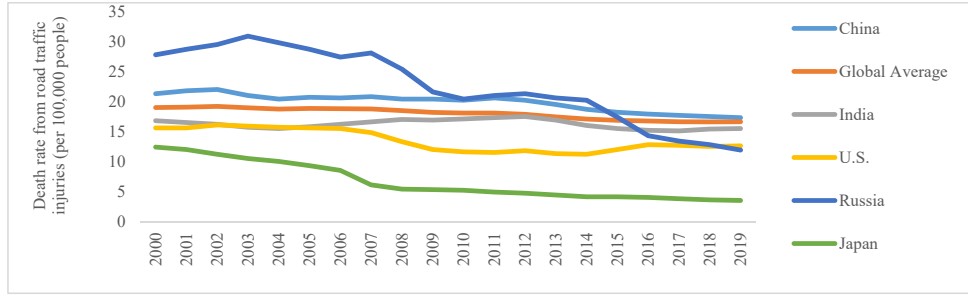

**Figure 3.** Comparison of death rates from road traffic injuries (per 100,000 people) in major countries. Source: data compiled from World Bank, 2021 [22].

### 4.3. Traffic Congestion

Megacities around the world suffer from traffic congestion (e.g., the average congestion time in London was 227 h, and that in Bogota was 272 h in 2018), especially emerging megacities in developing countries, as growth of urban populations and private motor vehicle ownership and insufficient infrastructure (such as roads, parking spaces, etc.) to serve automobiles leads to serious congestion problems. The structure of cities' circular and radial traffic networks leads to unnecessary urban traffic demand, which further exacerbates traffic congestion, especially during peak hours.

In terms of passengers, the time spent traveling to and from the destination is considered to be wasted or less valuable when the travel time exceeds the budget (about 65–70 min). Considering private car drivers, a 20-min longer commute is equivalent to a 19% pay cut [23]. Globally, the annual economic cost of traffic congestion (including delays and queuing time) is about USD 12 trillion, of which about USD 72 billion is in the United States (the cost of peak traffic congestion in the United States is about 13 cents/mile).

### 4.4. Worsening Traffic Pollution

The WHO (2016) found that around 92% of the global population lives in cities with excessive levels of air pollution. The transportation sector has been one of the main contributors to this issue [24]. It is suggested that the transportation sector in Asian cities is responsible for 80% of air pollution, and the problem will become more serious in the future with the growth of emerging Asian economies and the popularity of private vehicles. Among them, PM2.5 (whose sources mainly include the direct contribution of primary particulate matter emissions and the indirect contribution of secondary conversion of gaseous precursors such as sulfur dioxide ($SO_2$), nitrogen oxides (NOx), volatile organic compounds (VOCs) and ammonia ($NH_3$)) is one of the major air pollutants causing severe haze and harming human health, accounting for an estimated 7.3% of total global deaths [25]. As shown in Figure 4, the global annual average concentration of PM2.5 has been stable since 1990 at 4–5 times the WHO annual PM2.5 standard (which was tightened from <10 µg/m$^3$ to 5 µg/m$^3$ in the latest WHO Air Quality Guidelines in 2021). Among them, concentrations in Russia, the United States and Japan have slowly declined, but they are still higher than the WHO target. Air pollution in India, the world's second most populous country, is getting worse with the acceleration of industrialization and urbanization. Its average concentration of PM2.5 has fluctuated but increased overall since 1990, and it is much higher than the global average and more than 10 times that of the WHO. It caused 1 million deaths in 2015 alone [26].

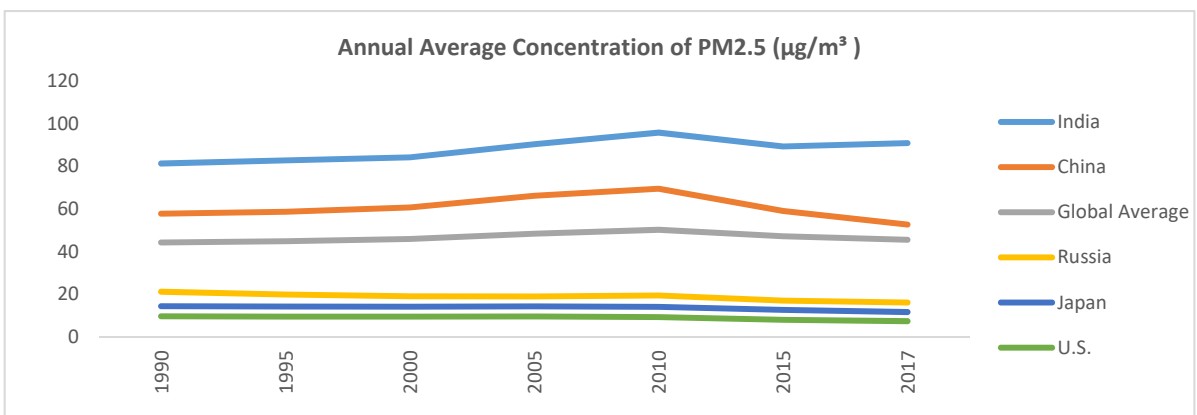

**Figure 4.** Comparison of annual average contraction of PM2.5 in major countries. Source: data compiled from World Bank, 2021.

*4.5. Climate Change*

GHG are gases that have the ability to absorb infrared radiation (net thermal energy) emitted by the Earth's surface and re-radiate it back to the Earth's surface, thereby contributing to the greenhouse effect. GHG emissions that cause climate change have increased 50-fold since the mid-1800s. The IPCC Fifth Assessment Report [27] pointed out that human activities, especially excessive use of fossil energy, are the main reason for the increasing concentration of GHG in the atmosphere due to the increase in $CO_2$ produced. The energy consumed and $CO_2$ produced by cities also exceed 67% and 70% of the global totals, respectively. Cities have been the largest contributors to GHG emissions. Transportation is not only the fastest-growing GHG emitter in the world (from 5.0Gt $CO_2$-eq in 1990 to 7.2Gt $CO_2$-eq in 2020, decreasing by more than 10% compared to 2019 due to lockdowns in response to COVID-19), but also the third largest source of $CO_2$ emissions after the power and industrial sectors (24% of global energy-related $CO_2$ emissions). Among them, road transportation accounts for 78.33% (road passenger transport and road freight transport account for 53.73% and 24.6%, respectively), shipping accounts for 11.43%, aviation accounts for 8.87%, and rails account for only 1.3%. Specifically, road transport by light-duty vehicles (LDVs), buses and minibuses as well as 2- and 3-wheeled vehicles accounts for 45.6%, 5.73% and 2.4%, respectively [28].

The Paris Agreement clarified the long-term goal of controlling the global average temperature rise to no more than 2 °C above pre-industrial levels by the end of this century and striving to control it within 1.5 °C. The annual global carbon emission in the next 10 years needs to be reduced by 7.6%, to achieve average net-zero $CO_2$ emissions by mid-century [29]. *Transforming our world: the 2030 Agenda for Sustainable Development* sets 17 Sustainable Development Goals (SDGs) and 169 sub-goals [30]. These frameworks provide a blueprint for a sustainable, low-carbon and fairer world. Both the Paris Agreement and the SDGs recognize the importance of developing sustainable energy systems to address the environmental, economic and social challenges posed by climate change [31]. *The Special Report on Global Warming of 1.5 °C* pointed out that according to the current monthly average temperature rise of 0.2 °C per decade, the global temperature rise may reach 1.5 °C in 2030 at the fastest. Once the critical point of 1.5 °C is exceeded, the frequency and intensity of climate disasters will increase significantly [32].

## 5. Trends in the Sustainable Development of Urban Transportation

Sustainability efforts initially focused on environmental issues and took place outside cities. However, Anders [33] believes that the city is the most suitable place to carry out such actions. In particular, the increase in global urbanization brings both challenges and opportunities to sustainable development [34]. Sustainable urban development must strike a balance between environmental protection, economic development and social well-being. That means reducing the ecological footprint (input of natural resources and output of waste) while improving urban livability (social amenities, health and well-being of individuals and communities) [3]. In 1992 (5 years after the release of *Our Common Future*), the United Nations' Earth Summit first recognized the role of transport in sustainable development. The European Union [35] also proposed the concept of sustainable mobility in the *EU Green Paper on the Impact of Transport on the Environment*, which is to ensure that the transport system meets the economic, social and environmental needs of society while minimizing adverse economic, social and environmental impacts. This was a direct response to the challenge posed by *Our Common Future*, and the concept of sustainable transport appeared on the international agenda for the first time. Sustainable transport has been defined both narrowly (i.e., focusing on resource depletion and environmental issues) and broadly (i.e., covering social well-being, economic growth and environmental protection) [36]. It is a system designed to provide safe, affordable and clean services for the movement of passengers and goods, whose negative effects can be maintained without compromising the ability of future generations to meet their basic needs [37].

In the decades following the end of World War II, massive increases in infrastructure such as roads were seen as a solution to traffic congestion and other urban problems. However, the Downs–Thomson paradox finds that the equilibrium speed of private cars on a road network depends on the average travel speed of people traveling from door to door using public transport. Although the demand elasticity of induced travel is still controversial, it has been confirmed that an increase in road investment can indeed lead to induced travel, so problems such as traffic congestion have not been alleviated [38]. Instead, greater road investment caused more damage to cities because of the need to demolish buildings or occupy public space [39]. The demand for passenger traffic in urbanized areas around the world will double by 2050. New technologies such as big data, artificial intelligence (AI) and the Internet of Things (IoT) have become the new direction of urban transportation development, promoting a major change in urban transport.

## 5.1. Transportation Demand Management

Traffic demand management originated in the 1970s and is considered an effective solution to alleviate urban traffic congestion [40]. It includes limitations on private car ownership and use, road pricing, attractive public transport systems and more. Gallego et al. [41] argue that restriction policies can effectively reduce traffic congestion and even rigid traffic demand in the short term. Therefore, the environment and safety have also been improved to some extent.

Mexico City implemented a one-day-a-week vehicle restriction policy as early as 1989 [42]. Beijing, as the first Chinese city to implement such a policy, adopted an odd-even policy for license plates during the 2008 Olympic Games (research showed that its traffic volume was reduced by 20–40%, and the driving speed increased by 10–20%). It has since shifted to a one-day-a-week vehicle restriction policy as a regular TDM strategy [43]. However, some residents travel illegally (for example, the proportion in Delhi, India, is as high as 20%), switch their travel to unrestricted time periods or buy new cars with alternating license plates [44]. Therefore, although travel restriction policies have a direct impact on travel demand, that impact is limited.

In 1994, Shanghai adopted Singapore's paid auction system and became the first city in China to implement a license control policy. In 2000, it introduced a public auction mechanism to allocate license plates. The number of private cars owned per 1000 persons in Shanghai is significantly lower than in other cities (for example, in 2019, the number of small and micro private cars owned per 1000 persons in Shanghai was 132, which is lower than the national average of 148 and far below the 217 in Beijing). Beijing, as the city with the highest rate of car ownership in China, has used a free lottery system since 2011 to slow down the rapid growth of motor vehicle ownership. For example, the total quota for passenger cars was 240,000 in 2011. Since 2014, the total quota has gradually decreased, but the proportion of new energy vehicles has gradually increased.

## 5.2. Electrification of Urban Transport

Land transport is one of the fastest-growing sectors amongst all energy sectors globally, with an annual growth of 1.7%. LDVs power systems mainly rely on gasoline and diesel. Vehicle exhaust is the main cause of air pollution: gasoline vehicles are the main source of CO and HC emissions; diesel vehicles are the main source of NOx emissions; and gas vehicles are the main source of PM emissions. The average fuel consumption of new internal combustion engine (ICE) vehicles has improved significantly but at a slower pace (LDVs fuel consumption decreased by 1.8% per annum in 2005–2016, but only 0.7% in 2016–2017). Its near-term and future improvements will not be sufficient to meet the level of deep decarbonization of the transport sector. Unless the sale of ICE vehicles is banned, LDVs will likely continue to feature conventional, hybrid and plug-in hybrid ICEs for the next 30 years.

Any emerging technology options for the transport sector, both domestically and internationally, will need to be combined with low-carbon and/or renewable energy to

have the potential to address the growing emissions challenges of the transport sector. It is not feasible to achieve the level of carbon reduction required in the transport sector with continuous reliance on fossil-fuel-based electricity. The alternative fuels for ICE vehicles are very important for reducing $CO_2$ and other emissions, but they are equally challenging. The natural gas-based fuels (compressed natural gas/CNG for commercial vehicles and light- and medium-duty vehicles as well as liquid natural gas/LNG for heavy-duty vehicles to replace gasoline and diesel, respectively) offer lower levels of pollution and mature technology. However, there are still limits on supply chain emissions, methane emissions and exhaust emissions. Biofuels are an important climate mitigation strategy for the transport sector under certain conditions. However, they are not competitive with existing fossil fuels due to their feedstock availability and cost, challenges in the production and distribution supply chain, etc. Ammonia as a carbon-neutral fuel and hydrogen carrier for fuel cells is more suitable for maritime transport. Other synthetic fuels can reduce GHG emissions without making significant changes to existing and new vehicle engines, but they will be mainly applied in the fields of aviation, shipping and long-distance land transportation due to being three times the price of conventional fossil fuels and offering potentially lower overall energy efficiency than electric vehicles.

It will be beneficial to improving air quality and promoting green jobs if electric vehicles are powered by sustainable energy sources. Current hybrid electric vehicles (HEVs) typically have smaller batteries compared to battery electric vehicles (BEVs) and therefore can reduce emissions by up to 30% compared to ICEVs (but are mainly dependent on fuel). Since HEVs rely on combustion as their primary energy conversion process, they offer limited mitigation opportunities and are a suitable interim solution. Due to their dual operation characteristics, the lifetime GHG emission intensity of plug-in hybrid electric vehicles (PHEVs) can be expected to be between those of ICEVs and BEVs of similar size and performance.

It is only possible to limit the global temperature rise to 2 °C by 2030 if at least 20% of road transport vehicles (about 300 million vehicles) are electrified [45]. It has been found that more than 60 billion tons of $CO_2$ emissions could be avoided by 2050 if 60% of road transport vehicles were electrified [46].

*5.3. Intelligent Transportation System*

Schumpeter [47] explained the innovation of business and technological systems during economic collapse on the basis of the induction of economic waves by the Russian economist Kondratieff [48]. Freeman and Soete [49] further extended it to a long-cycle theory of economic development. Specifically, the first wave of innovation significantly reduced production and transportation costs by integrating new technologies and enabling a shift from artisanal to industrial production; the second wave of innovation, marked by the age of steam, promoted the development of railway transportation and enabled long-distance movement of people and goods; the third wave is the age of electricity, which promoted the development of urban public transportation and the automobile industry and expands the mobility of passengers and goods; the fourth innovation wave promotes the globalization of mobility; the fifth innovation wave is based on information and communication technologies and networks. Hargroves and Smith [50] predict a sixth wave of innovation related to sustainability and digitalization. Batty [51] believes that smart city technology is the focus.

The term smart city originated in the mid-19th century to describe new cities in the American West. In the 1990s, with the development of the Smart Growth movement, smart cities were proposed to achieve sustainable urbanization [52,53]. Although some scientific research and government agencies have proposed frameworks for smart cities, the most widely used one is the Smart City Wheel proposed by the European Union, which covers most but not all areas of smart cities. It includes: (1) smart economy; (2) smart environment; (3) smart governance; (4) smart living; (5) smart people; and (6) smart mobility. Smart transportation refers to the process and practice of integrating

information and communication technologies (information and communications/ICTs) and other cutting-edge technologies (IOT, etc.) into transportation [54].

- Shared Mobility

Shared mobility is the fastest-growing area of the sharing economy, with Asia and Europe being the world's largest car-sharing regions (accounting for about 43% and 37% of the world's sharing of cars, respectively) [55]. Shared mobility provides accessibility for marginalized groups (e.g., the elderly, the disabled, etc.) and when integrated with local rapid transit corridors can greatly reduce the overall demand for private cars. However, if shared mobility leads to an overall shift in transportation, such as an increase in private motorized travel (such as Schaller's 2018 study found that shared vehicles such as Uber led to increased vehicle mileage) and reductions in public transport or walking, it may generate higher demand for private travel and lead to more negative environmental impacts [56].

- Customized Buses

For specific customers (especially commuters), customized buses provide high-quality, personalized, flexible and demand-responsive public transportation by using the internet, telephones and other online information platform-interactive services to aggregate similar travel needs. This is a form of transportation that is somewhere between a regular bus and a private car. Compared to traditional public transportation, a customized bus is more reliable and comfortable. It also helps reduce traffic congestion and other environmental problems compared to private cars.

- Mobility as a Service/MaaS

The business model of MaaS is stimulated by the smartphone market [57]. With the powerful functionality (for example, transportation operators can provide travelers with real-time travel plans through smartphones; travelers can provide information to service platforms through the location awareness functions of smartphones, etc.) and popularization of smartphones, more and more young people tend to take public transportation, so as to use travel time to work or socialize and maximize the value of their travel time. MaaS is a mobile platform based on potential applications that can integrate different transportation modes (such as shared transportation, taxis, ride-hailing and bicycles) into a single user interface. In this way, users will receive information about different options for travel and can choose an appropriate combination according to time and cost; they can then make a one-time payment for the services they will use. Therefore, MaaS is not a new invention but an evolution. MaaS is based on the basic concept of realizing the sustainable development of transportation, and it provides users with one-stop service through demand-response, "door-to-door", effective integration of travel modes and payment facilitation [58]. MaaS will provide more citizens with the opportunity to meet their travel needs without buying a car, and the travel mode is diversified/integrated/personalized. In this way, private motorized travel methods are minimized, green travel is encouraged, and lower carbon emissions are realized.

*5.4. Transit-Oriented Development/TOD*

In the 1920s, with the increase of wealth, the development of cities and the popularity of automobiles, urban dwellers in United States began to tend to buy larger residential lots and larger houses in single-function suburbs, which have been derogatorily termed "McMansions". Suburban development is gradually outpacing city center development. In the 1950s, the phenomenon of "counter-urbanization" appeared, with urban sprawl development, serious functional zoning and surging private motorized travel.

Under the concept of sustainable new urbanism, Calthorpe [59] in his book *The Next American Metropolis: Ecology, Community, and the American Dream* formally proposed the concept of TOD, which is to combine transit and land use through concentrated development around transit stations (train stations, light rail stations or bus stops). TOD concentrates land use around bus stations or in bus corridors. The density of cities close to stations is

higher, and it gradually decreases to a lower density outside a radius of 800 m, but it is still higher than typical automobile cities such as those commonly found in the United States. It also enables people-centered urban transport systems by removing unsustainable vehicle infrastructure and prohibiting further development (such as the demolition of the Cheonggyecheon Expressway in Seoul in 2002). Transportation demand is to be reduced through increased urban density and mixed land use. Future urban development is using new rail lines (light rail and metro) as anchors (e.g., along the TOD corridor of Curitiba BRT, Brazil). The first- and last-mile needs of rail transit users are supported through walking, biking, bus shuttles, taxis, car sharing, bike sharing and all available modes of transportation. Thereby, a sustainable urban transport system is provided to meet the daily travel needs of residents and encourage sustainable urban growth rather than disorderly urban development.

## 6. Conclusions

Aristotle once said that human beings "gathered in cities in order to live, and stayed in cities in order to live better". As a material space, the city accommodates all kinds of human production and creative activities. The global urban population will reach 5 billion by 2030, and the average urban expansion rate is twice that of its population. The global urban land cover will increase by 1.2 million square kilometers by 2030, reaching almost three times the level of 2000. The global number of cars will also triple by 2050. In developing countries, the growth rate of car ownership is the fastest. Cars provide residents with a convenient, fast and private way to obtain goods, services and activities, and in some countries they have become symbols of identity and social status. At the same time, the automotive industry has become one of the major industrial sectors of the global economy since the late 19th century, and it has strong economic ties with other industries, which also magnifies its economic importance. However, a private vehicle-based approach to providing transportation to cities has many insurmountable shortcomings, as it has a range of adverse economic, environmental and social consequences. For example: urban sprawl, long commuting distances, traffic congestion, property damage, casualties from road traffic accidents, damage to physical and mental health, excessive energy consumption, environmental pollution and climate change. In short, it has a negative impact on livability, safety, resilience and sustainability, which are the core goals of SDG11 and are at the heart of environmentally sustainable transport as articulated by the Bangkok Declaration.

TDM measures are taken to restrict the ownership and use of private cars. However, restrictions on the use and ownership of private cars and innovations in fuels and technology will all fall short of expectations due to increased traffic. Electric vehicles powered by low-carbon energy and/or renewable energy have the greatest decarbonization potential in land transportation. Sustainable biomass fuels can provide additional emission reduction benefits for land transport in the short to medium term. Shared and intelligent urban transportation systems make it convenient for residents to adopt green travel methods and public transportation. The TOD development model based on rail transit encourages mixed and compact use of land around traffic nodes and along traffic corridors, thereby generating two-way traffic flows and avoiding commuter tidal traffic. It also creates a people-oriented urban center that achieves accessibility for residents, including those with disadvantaged transportation, while also improving local economic development.

Global traffic activity levels increased by 73% between 2000 and 2018 with a continuous growth in the demand for freight and passenger services in the long term. As the world's largest emerging economy, China surpassed United States as a global auto giant in 2009 and became the world's largest energy producer and consumer, accounting for 13% of the global total. It has been the world's largest carbon emitter since 2007 and accounted for 28% of the global total in 2019, while the 100 countries with the lowest global emissions accounted for less than 3%. In its Nationally Determined Contributions (NDCs), China reiterated that it would reach peak $CO_2$ emissions around 2030 and strive to achieve it as soon as possible. Discussing the characteristics and trends of urban transportation and

realizing carbon reduction in the transportation sector are important measures for China to achieve its carbon peaking and carbon neutrality goals.

**Author Contributions:** Conceptualization, formal analysis, writing, review and editing, funding acquisition, Y.G. Data collection and referencing, J.Z. All authors have read and agreed to the published version of the manuscript.

**Funding:** This research was funded by the Education Department of Henan Province grant number 20B630021 and Zhengzhou University grant number 32220525.

**Conflicts of Interest:** The authors declare no conflict of interest.

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
