# Peer review of "Characteristics, Impacts and Trends of Urban Transportation"

_encyclopedia, doi:10.3390/encyclopedia2020078_

Round 1

Reviewer 1 Report

  1. The manuscript presents good information. Therefore, I suggest to the authors review the information and the connection with the information. Some suggestions to improve the manuscript are below
  2. The characteristics, impacts, and trends are weakly informed. Sam
  3. What is the intent of section 2? The title is so generic, and the content is so weak. Focus only on the characteristics of urban transportation. However, urban transportation is more the public transportation. I suggest that the authors define the concept of urban transportation well in the paper.
  4. Traffic congestion is a negative impact. The authors do not need to highlight the "worsening". Same comment for the title of subsection 4.5
  5. traffic pollution is closely related to climate change. Think about the possibility to merge these subsections.
  6. Check the subsection of Transportation Demand Management (bullets information). The authors could write it in one section without bullets.
  7. What means "Intelligentization of Urban Transportation"? Is it related to the intelligent transportation system? 
  8. The conclusion must be improved.

Author Response

Thanks to all your comments. I have accepted all comments and made changes accordingly. All revisions are shown in the word file by track change. 

Reviewer 2 Report

The proposed entry is adapted to a topic of special interest, for a large number of urban policies, such as public transport. The text is well written and consistent. The authors manage to provide a basic review of the state of the art and trends on urban public transport. However, some issues should be addressed to improve the final result of the manuscript:

  • Line 43: "The United States has a typical "automobile city” fabric". This is true for much of the urban space in American cities, but not for urban centers, which were previously projected.
  • Line 76: Examples of public service-oriented transport services exist much earlier in history.
  • After line 83: It would be appropriate to anticipate the organization of the following contents, as a remainder.
  • 3.1. Regular bus: what are the advantages and disadvantages compared to other modes of urban public transport? Are they distributed equally in any city or are there regional differences? Trolleybuses could be incorporated.
  • Line 110: it seems that there are 5 cities between Africa and Oceania, when in fact there are 5 on each continent.
  • Lines 120-122: if it only represents 2.2% overall, the role of the normal bus should be reinforced in the text compared to the BRT systems, since these are much more important in the mobility of cities.
  • 3.3. Urban Rail Transit: it could be better differentiated and explained between metro (underground) and other types of urban railways (trams, suburban railways, mixed systems, etc.).
  • 4.2. Increase in traffic accidentes: it would be useful to clarify whether these accident figures are total by country, that is, do they include urban and interurban traffic? Is there information on urban transport accidents?
  • Line 196: SO2
  • Line 197: NH3
  • Line 227: CO2
  • 4.5. Climate change: these numbers seem to be general data, do they include aviation? Are there estimates showing the percentage or total of GHC emissions contributed by each mode of urban transport?
  • Lines 260-261: at the same time, railway lines in European countries and in America were dismantled through the initiative of numerous governments.

Author Response

(The authors gave the same response as above.)

Reviewer 3 Report

Review:

Characteristics, Impacts and Trends of Urban Public Transportation

I have read this article and commented that:

  1. The articles presented are still uninteresting, not new.
  2. The author have mentioned, how to balance how to balance the relationship between people’s growing demand for private motorization with the development of urbanization, which has not yet found a summary of such questions.
  3. The content of the article is still inconsistent. Authors should display a flow of properties, effects, causes, trends, and approaches to provide a solution to the problem.

Author Response

(The authors gave the same response as above.)

Reviewer 4 Report

The article is well documented and clear presented. The article has an adequate theoretical basis, relevant information and analysis, good partial (in the article) and final (in the conclusion) conclusions.

Author Response

(The authors gave the same response as above.)

Reviewer 5 Report

In line with economic development and urbanization, the entry discusses the characteristics, impacts, and trends of urban public transportation. In general, the paper an interesting and well-thought topic. 

Only minor corrections are needed to further improve the paper:

  1. The entry is submitted to the Topical Collection "Encyclopedia of Engineering" with the following categories: History of Mechanics, Civil Engineering, Mechanical and Aerospace Engineering, and Nanoengineering. The discussion should be anchored to any of these topics, e.g. civil engineering.
  2. Urban Public vs. Private Transportation should be defined and distinguished.
  3. Should Section 3 be: "Characteristics of Public Urban Transport" only, as the Impacts and Trends are discussed in the next sections?
  4. Conclusions should be divided into paragraphs by topic. Highlight the main results for each: characteristics, impacts, trends; and mention them.
  5. Figures and tables should be improved. Make the fonts larger to make them readable, while increasing the quality of the figures. Also, figure and tables should be self-explanatory, avoid using acronyms as much as possible, otherwise, define them as notes.
  6. Minor issues on capitalization of subheadings and uniformity in using (Oxford) comma. Also the "source/resource"? in the tables/figures.
  7. Use a uniform Reference format. Follow the Author Guidelines.

Author Response

(The authors gave the same response as above.)

Round 2

Reviewer 1 Report

I thank the authors address my suggestions to improve the paper.

Author Response

Thank you very much for your valuable comments.

Reviewer 2 Report

The authors have addressed and incorporated a portion of the suggested recommendations. However, in the opinion of this reviewer there are two outstanding issues:
1. Why has the text now been expanded with China-specific information? This is not reflected in either the title or the abstract; If this change is made, then shouldn't I also pay more attention to other countries in order to compare and obtain a complete and critical view?
2. Different comments have not been addressed, please indicate that they are rejected (expressing the reasons) or that they are really accepted and incorporated. E.g.

3.1. Regular bus: what are the advantages and
disadvantages compared to other modes of urban public
transport? Are they distributed equally in any city or are
there regional difference? Trolleybuses could be
incorporated.

Line 260-261: at the same time, railway lines in European
countries and in America were dismantled through the
initiative of numerous governments.

Author Response

Thank you very much for your valuable comments. Please find the responses in the attachment.

Reviewer 3 Report

-

Author Response

(The authors gave the same response as above.)

Reviewer 5 Report

All comments and suggestions were carefully addressed. The revised manuscript has significantly been improved.

Author Response

(The authors gave the same response as above.)

Round 3

Reviewer 2 Report

The authors have addressed and incorporated review comments.